# Brachycephaly, Ear Anatomy, and Co—Does Size Matter? A Retrospective Study on the Influence of Size-Dependent Features Regarding Canine Otitis Externa

**DOI:** 10.3390/ani15070933

**Published:** 2025-03-24

**Authors:** Peter Christian Ponn, Andrea Tipold, Sandra Goericke-Pesch, Andrea Vanessa Volk

**Affiliations:** 1Department of Small Animal Medicine and Surgery, University of Veterinary Medicine Hannover, Foundation, 30559 Hannover, Germany; andrea.tipold@tiho-hannover.de (A.T.); andrea.volk@tiho-hannover.de (A.V.V.); 2AniCura Recklinghausen—Small Animal Clinic, 45659 Recklinghausen, Germany; 3Unit for Reproductive Medicine—Clinic for Small Animals, University of Veterinary Medicine Hannover, Foundation, 30559 Hannover, Germany; sandra.goericke-pesch@tiho-hannover.de

**Keywords:** dermatology, canine, otitis externa, pinna formation, reproductive medicine, brachycephaly, prophylaxis, retrospective

## Abstract

Canine Otitis externa is one of the most common presentations in veterinary practice. Therefore, many studies have been conducted to analyze predispositions. This study took a different approach in evaluating potential protective, mostly size-dependent, features. It was found that Rhodesian Ridgebacks, intact female dogs, semi-erect pinna formation, large-sized breeds, and non-brachycephalic breeds had a decreased risk for developing Otitis externa. On the other hand, medium-sized dogs, erect pinna formation, neutered (male and female) dogs, Cocker Spaniels, Pugs, French Bulldogs, and brachycephalic breeds, in general, tend to develop Otitis more frequently. All these factors can be taken into consideration as criteria before making certain decisions regarding which dogs to choose for breeding and if a dog should be neutered or not. Furthermore, the predisposition for brachycephalic dogs adds one more health issue to their already long list.

## 1. Introduction

Size, by definition describing how large or small an individual or an organ is [1], has an enormous influence on many processes in various fields of veterinary medicine. From actual body size in neonatology (e.g., dystocia regarding the body sizes of the unborn litter [2]) or oncology (e.g., the prevalence of osteosarcoma in large breed dogs [3,4]) to the occurrence of osteoarthritis in larger breeds and obese patients [5,6,7], size has a direct influence on many pathophysiologies [8].

Several papers haven been published on the relationship between ear size and canine Otitis externa [9,10,11,12], and potential correlations have been suggested. Otitis externa (OE) describes a common inflammatory condition ranging from pruritus to severe pain leading to an impairment in the quality of life. Etiologies of OE can be numerous and are classified in different categories via the PSPP system (primary, secondary, predisposing, and perpetuating factors) [13].

Most of these studies aimed to evaluate potential predispositions regarding canine OE. Many anatomical conditions, some not size-dependent like the pinna formation [12,14], as well as those that are size-dependent like the diameter of the ear canal [10], have been shown to influence the prevalence of OE. Another size-dependent feature affecting a very specific group of dogs is brachycephaly. These dogs suffer from numerous diseases due to their altered skull (and body) morphology; frequently, severe ear disease is diagnosed [9,15,16].

Although not definitively confirmed, general factors such as gender, neuter status, and weight may also influence the risk for OE [12,14,17,18]. In cases of gender and neuter status, various studies have reported different conclusions regarding ear diseases [14,17,18]. Other studies, nonetheless, have confirmed that sexual hormones alter the constitution of the skin (from dry to oily) [12]. Furthermore, it is known that neutering can lead to obesity in pets [19].

While there is still a lack of data regarding obesity and its potential influence on OE, it is known that adipose tissue functions as an endocrine organ itself. In the human field, metabolic syndrome is an endocrine disease that is linked to being overweight and obese [20]. In equines, a similar definition is used for risk factors that increase the risk for endocrinopathic laminitis. Although exceptions occur, this condition is mostly associated with obesity [21]. Besides that, obesity also strengthens proinflammatory processes and increases the risk for cardiovascular diseases and other endocrinopathies like diabetes [8,22].

While most of the other factors are not changeable during a dog’s life, being overweight could be counteracted by weight management. This means obesity is a factor that could be addressed proactively by owners as well as veterinarians.

Contrary to the published data described above, factors associated with a decreased risk of developing OE have not been previously investigated. Therefore, recently, a study was conducted to evaluate potential features that minimized the risk for OE [23]. This current follow-up-study was intended to follow the new approach, and it aimed to investigate both size-dependent and general phenotypic features and their influence on the presence of canine OE, again directed toward potential protective factors.

## 2. Materials and Methods

### 2.1. Study Population

The inclusion criteria for this retrospective study were body type (small–medium–large size), pinna formation (pendulous, V-shaped, semi-erect, and erect), and brachycephaly. For brachycephaly, four breeds (Pugs, Boxers, Chihuahuas, French Bulldogs) were chosen as seen in Figure 1. The collection of cases diagnosed with OE was carried out in the patient management system (EasyVet^®^; VetZ GmbH; Isernhagen, Germany) of the department of Small Animal Medicine and Surgery of the University for Veterinary Medicine Hannover between 1 January 2019 and 31 July 2022.

As a control group, dogs of the Unit for Reproductive Medicine of the university hospital were chosen, matching in size/pinna formation and breeds, and were presented for routine checks during the same time period as the study group. The control group was selected from the reproduction clinic to minimize the incidence of comorbidities in this population. Dogs in the control group, which were diagnosed with OE at some point, were excluded from this study as seen in Figure 2.

### 2.2. Signalment

Cases were analyzed based on the following criteria: presence or absence of OE, breed, sex and neuter status, body type, pinna formation, overweight, and brachycephaly. A more detailed rundown on the selection process is shown in Figure 2.

The diagnosis of OE was provided by the vet on duty or retrospectively based upon at least two of the following criteria: clinical signs (e.g., inflammation, hyperemia, ulceration of the ear canal with pruritus, ceruminous discharges) and/or cytological abnormalities (e.g., inflammatory cells or an increased amount of bacteria/yeast) in the record.

Breeds were chosen by the criteria of pinna formation and body type. The pinna formation was divided into pendulous, V-shaped, semi-erect, and erect ears, while body type was grouped in small-sized, medium-sized, and large-sized breeds according to VDH criteria [24]. At least one breed of every body type and pinna formation was selected and included in the study. Breeds that were previously associated with brachycephaly in the literature were chosen by the same criteria and added to the existing pool [10,25,26,27]. For pendulous and semi-erect ears, no brachycephalic breed was chosen. Regarding pendulous ears, Cocker Spaniels sometimes count as brachycephalic breeds but were not included in this study’s brachycephalic pool as they can express mesocephalic features as well [28]. Furthermore, for the group of semi-erect ears, no fitting brachycephalic breed could be determined. Breeds of the same breed type, with similar body types and pinna formations, were pooled into one breed group to ensure a representative sample size. Eventually, the 18 breeds shown in Figure 1 were chosen for this study.

Regarding breed analysis, all brachycephalic breeds were united and compared to non-brachycephalic breeds to determine a statistical correlation between these breeds and OE.

Furthermore, gender and neuter status were categorized into four groups (intact female, neutered female, intact male, neutered male).

Body weights were grouped as “overweight” or “not-overweight”. Overweight was chosen when the individual weight was above the breed‘s weight range according to the American Kennel Club^®^ (American Kennel Club; Raleigh, NC, USA) [29].

### 2.3. Clinical Examination

In the study group, cases were recruited by the diagnosis of OE by the dermatology group or out-of-hours services. A complete physical and dermatological examination including an otoscopy needed to be performed as well as cytological samples collected, processed, and analyzed as published previously [30,31].

The control group, presented to the Unit for Reproductive Medicine, underwent a complete physical and reproductive system examination. Only dogs without any aural abnormalities or another clinical sign of OE were included.

### 2.4. Statistical Analysis

As all data sets contained nominal data, no tests for normal distribution were necessary.

The Pearson–Chi^2^ test and Fisher’s exact test were applied to assess the associations between two categorical variables. *p*-values of <0.05 were considered statistically significant. Furthermore, statistically significant results underwent post hoc testing—Bonferroni correction.

Analyses were conducted using the IBM SPSS Statistics 27.0.1.0 ™ (IBM; Armonk; New York, NY, USA) software, while Bonferroni correction was applied using Microsoft Excel™ (Microsoft; Redmond, DC, USA). Bivariate analysis was selected to evaluate significance between each factor and the presence of OE individually.

## 3. Results

### 3.1. Study Population

Out of 868 cases, 180 (20.7%) showed clinical or cytological signs of OE. The group of affected patients consisted of 29 intact female (16.1%), 44 neutered female (24.5%), 62 intact male (34.4%), and 45 neutered male (25.0%) dogs.

Four breeds were overrepresented: French Bulldogs (7.94% of the total study population), Rhodesian Ridgebacks (12.9%), Shepherd dogs (13.94%), and Retriever dogs (22.46%).

### 3.2. Body Size

A significant association was evident between body size and presentation of OE (Pearson–Chi^2^ *p* < 0.001|Fisher’s exact test *p* < 0.001). Figure 3 shows a graphical presentation of the distribution of the breeds grouped by their body size and presence of OE.

After post hoc testing using Bonferroni correction, significance was determined in medium-sized breeds (*p* = 0.000014 in OE present) and large-sized breeds (*p* = 0.000023 in OE, non-present).

### 3.3. Pinna Formation

A significant association was found between pinna formation and the development of OE (Pearson–Chi^2^ *p* < 0.003|Fisher’s exact test *p* < 0.002). Adjusted residuals showed that the responsible features were erect pinna regarding the presence of OE and semi-erect pinna regarding the non-presence of OE. After post hoc testing using Bonferroni correction, it was found that significance remained in both cases: erect ears (*p* = 0.003574) and semi-erect ears (*p* = 0.003595), as seen in Figure 4.

### 3.4. Sex and Neuter Status

For further analysis of sex and neuter status, the 27 cases without assigned gender were excluded in this regard. Out of the 440 intact females, 29 (6.6%) were presented with OE while 411 (93.4%) were not. In the neutered female group with 67 cases, 44 (65.7%) were diagnosed with OE and 23 (34.3%) were not (Figure 5).

The 281 intact male subjects were divided into 62 (22.1%) dogs with OE and 219 (77.9%) dogs without signs of OE. Out of the 53 neutered male population, 45 (84.9%) were presented with OE and 8 (15.1%) without.

There were statistically significant associations found between the presence of OE and gender and neuter status (Pearson–Chi^2^ *p* < 0.001|Fisher’s exact test *p* < 0.001), even after Bonferroni correction (*p* < 0.00001): intact females with positive residuals in the non-present-OE group and both genders neutered with positive residuals in the present-OE group.

### 3.5. Breeds

The statistical significance for the association between the presence of OE and breeds (Pearson–Chi^2^ *p* < 0.001) is shown in Table 1. A graphical representation of all breeds and their total numbers is shown in Figure 6, while Figure 7 displays the percentage distribution of the presence of OE grouped by breeds.

Subsequently, a post hoc test was applied. Every case with an adjusted residual > 1.9 was included. Nine breeds were included in this test: Appenzell Mountaindog with positive residuals in the present group, Greyhound in the non-present, Pug in the present, Rhodesian Ridgeback in the non-present, Setter dog in the non-present, Cocker Spaniel in the present, Collie in the non-present, Dachshund in the non-present, and French Bulldog in the present.

After performing Bonferroni correction, significance was still shown in four breeds: Pug (*p* = 0.000356), French Bulldog (*p* = 0.000001), Rhodesian Ridgeback (*p* = 0.000005), Cocker Spaniel (*p* = 0.00000000025).

### 3.6. Overweight

Breeds, summarized in groups and subjects, which had no weight registered, were excluded from the analysis, so a total amount of 304 cases remained. The distribution of these 304 cases can be seen in Figure 8.

Although a slightly higher number of overweight cases presented with OE, no statistically significant association between overweight and presence of OE was determined (Pearson–Chi^2^ *p* = 0.475|Fisher’s exact test *p* = 0.520|OR = 1.2 CI = 0.724–2.000).

### 3.7. Brachycephalic Breeds

A significant association between OE and brachycephalic breeds was shown (Pearson–Chi^2^ *p* < 0.001|Fisher’s exact test *p* < 0.001|OR = 2.083 CI = 1.434–3.025). Out of 175 brachycephalic subjects, nearly a third with 55 patients (31.4%) were diagnosed with OE, as seen in Figure 9.

After Bonferroni correction, the significance for OE in brachycephalics versus significance for non-OE in non-brachycephalics (*p* = 0.000095) was apparent, as seen in Figure 10.

## 4. Discussion

This study was conducted to evaluate potential protective factors for OE with a focus on physical criteria like size, pinna shape, gender, neuter status, as well as body weight. These criteria were chosen due to the fact that the conformation (including length) of the ear canal has an influence on the presence of OE. In addition to some general factors like breed and gender, features that may alter the dimensions of the ear canal were chosen as well [12].

The breed showing the greatest significance, even after Bonferroni correction, was the Rhodesian Ridgeback, which had a reduced risk of OE. A similar result was achieved in another study [23]; while the statistical approach was similar, the recent study included a different study and control population. Additionally, the Border Collie missed statistical significance in our data set, being a potential breed with a low risk of developing OE, as shown in other studies as well [12].

On the contrary, the Cocker Spaniel showed a statistically significant association with OE, which is consistent with a previous epidemiological study [32]. Another study concluded that dogs with hairy, pendulous ears, like Cocker Spaniels, tend to have a higher risk of developing OE [14]. These phenotypic features may disrupt airflow and increase heat retention inside the ear canal. Furthermore, Cocker Spaniels tend to have an increased number of ceruminous glands in their ear canals [33,34]. Such breed-specific conditions of the ear anatomy are one of the reasons why breeds can alliterate the risk of OE in both ways. Regarding breeds and their representation in this study, it should be mentioned that a potential bias may have occurred due to choosing dogs from the Unit for Reproductive Medicine. It should be borne in mind that these patients may vary depending on the popularity of certain breeds, as seen in the four overrepresented ones. This could be the reason why Shepherds and Retrievers, which are often associated with OE, did not show significance in this study [12,35,36,37]. Nonetheless, French Bulldogs, even though overrepresented, still showed an increased risk of suffering from OE.

As pendulous ears are commonly known to increase the risk of OE, in this study, different pinna formations were analyzed for a potential protective influence. It has been postulated that the ventilation of the ear canal reduces due to the closure of the dropped pinnae [17,38]. In this study, however, no significance could be found regarding pendulous ears. On the contrary, erect ears seemed to have an increased risk for OE, while semi-erect ears showed reduced risks. The only difference in these two pinna formations would be the slightly dropped pinnal upper third in semi-erect ears. This anatomical difference may protect the outer ear canal from environmental influence while remaining beneficial for ventilation due to the erect base of the pinna. The result regarding erect ears did not match previous studies [14]. A possible explanation for our results may be the overrepresentation (*n* = 69) of French Bulldogs. These dogs showed significance breed-wise, having erect ears. Unfortunately, the contradictory study did not include a complete list of breeds evaluated [14]. However, regarding the growing popularity of French Bulldogs and other brachycephalic breeds in Western countries during the last 12 years, their proportion in the former study group may not have been high, as the study was conducted in 1986. The number of registered French Bulldogs in the UK rose from 2771 dogs in 2011 to 36,785 dogs at the end of 2018, right before the starting point of this study [39]. The commonly assumed protective properties of erect ears could be seen in their semi-erect formation. Finally, it should be noted that a potential statistical influence for the significance in semi-erect ears lies in the absence of brachycephalic breeds in this pinna formation group. Nonetheless, further studies regarding potential differences between semi-erect and erect ears should be conducted.

In the current study, brachycephalic breeds, in general, showed statistical significance in the presence of OE. In a recent study, a higher frequency of sterile middle-ear effusion was shown in brachycephalic dogs, which may contribute to the development of OE [9,10]. Furthermore, the skull shape of brachycephalic breeds was shown to be associated with narrow ear canals in their horizontal parts [10]. This and the additional folding of their ear canals lead to reduced airflow and more heat radiation. Both are factors that support and prolong inflammatory processes [12]. Another study that investigated the health status of brachycephalic dogs in the UK found a significantly higher proportion of brachycephalics with OE as part of a multivariable analysis. However, a univariable analysis did not show significance [40]. In summary, based on the study in the UK and the present one, a significant correlation between OE and brachycephalic breeds could be established in both multivariable and univariable analyses.

Regarding sex and neuter status, previous studies have shown contradicting results. Some studies found a significantly higher risk in male dogs [14,17], while other studies did not find any significance according to gender distribution [18]. In our data, intact female dogs showed significantly less frequent inflammation of the outer ear canal (see Figure 3). However, both genders tend to have a statistically higher risk for OE once neutered. This may be due to the missing impact of steroid hormones on the skin, particularly the glands in the ear canal. Studies investigating the effects of sexual hormones on the skin barrier and immune response [41] have shown that estrogen and testosterone have effects on the constitution of the skin [12]. While estrogen leads to a drier skin, testosterone causes the skin to become oilier [12]. Furthermore, several studies have detailed an association between neutered dogs and being overweight [42]. A direct connection between being overweight and the presence of OE has yet to be established, but recent studies have suggested a potential influence of obesity on OE [23]. Even though, to the best of the authors’ knowledge, unfortunately, no studies have been conducted to investigate the potential influence of sexual hormones on OE, associations were found between gender and allergic dermatitis, one of the main primary causes of Otitis [12,18,43,44]. The studies investigated potential gender-related factors in canine atopic dermatitides and general allergic dermatitides in companion animals [45,46,47], as it has been shown in human medicine. The prevalence of atopic dermatitis is higher in women than in men [41]. While some studies yielded interesting results, like a higher risk of developing canine atopic dermatitis in neutered dogs in a study that involved 90,090 dogs, in summary, no meaningful significance could be determined due to limited data [45]. Further studies to evaluate the influence of sexual hormones on OE and dermatopathies, in general, would help to understand this phenomenon.

Regarding body size, the results of this study showed, on the one hand, that medium-sized dogs tend to have an increased risk of developing OE. This may be due to the fact that with an increased body size, the length of the ear canal increases proportionally and with this the risk of OE. But this raises the question of why large-sized dogs, on the other hand, showed a decreased risk for OE. It has been theorized that ventilation also depends on the diameter of the ear canal and plays a role in the etiology of OE [9,10,11]. Thus, this factor is worthy of further investigation. A larger diameter of the ear canal could lead to a better ventilation and so a decreased risk for OE, levelling out the increase in risk due to the longer ear canal. Further studies should be conducted to determine whether a larger diameter of the ear canal may be associated with a protective influence regarding OE, even though a small diameter may not support the development of OE. In these studies, breed-specific conditions, like the number of ceruminal glands, should also be taken into account.

In previous studies, obesity in dogs has been shown to be commonly associated with metabolic or endocrine abnormalities [8]. Even though in this study no significance was found, a slightly larger percentage of the dogs suffering from OE were overweight. Furthermore, in human and equine medicine, obesity is a huge factor in the pathogenesis of the “metabolic syndrome” [21,22]. Besides being a passive energy storage, adipose tissue is also known to function as an endocrine organ. It forms an integrated unit consisting of adipocytes, connective tissue, nerve tissue, immune cells, and stromovascular cells [20]. Adipokines like leptin and adiponectin, together with numerous other endocrine-active peptides, are secreted by adipocytes. As leptin’s structure is similar to IL-6-cytokines, it is assumed to have proinflammatory attributes [8]. Additionally, it has been shown in Labrador Retrievers that the leptin concentration is positively associated with a higher body condition score, while another study suggested the presence of immunological dysregulations in obese dogs [48,49]. Furthermore, it is commonly known that neutered or spayed dogs tend to become overweight [42], although recent studies have concluded that only the neutering of male dogs statistically increases the risk of obesity [19]. Regarding endocrine abnormalities, two studies in the early 2000s investigated, besides other disorders, the concurrent presence of OE in dogs with diabetes mellitus. One found the presence of OE or dermatitis in the study population with a prevalence of 16% [50]. In the following study, which focused on dermatologic disorders in diabetic patients, OE ranked second with 58% [51]. Furthermore, another study showed that OE associated with rod-shaped bacteria is significantly more frequent in dogs with endocrinopathies [18]. As information about the influence of obesity and its precise involvement in diseases in veterinary medicine is limited, this study also investigated a possible correlation between being overweight and OE, although no statistical significance was detected in our data set.

## 5. Conclusions

In conclusion, the results of this study show, for the first time, a lower susceptibility of larger breeds in comparison to medium- and small-sized breeds, possibly due to the larger diameter of the ear canal but definitively due to a longer muzzle. Furthermore, it is also the first time that a bivariate analysis is determining an increased risk for brachycephalic breeds, adding one more disease to the already extended list of health problems in these breeds. Regarding pinna formation, the results of this study show contradicting results to previous studies for erect ears. While the statistical approach may have influenced the results, further studies should be conducted to evaluate pinna formation as a potential factor influencing susceptibility to OE.

Regarding general factors, gender and neutering may influence the prevalence of OE based on the results of this study. Furthermore, it is shown, once again, that specific breeds tend to have a low risk (Rhodesian Ridgeback) for developing OE while others seem to have an increased risk (Pugs, French Bulldogs, Cocker Spaniel). Even though none of these factors can be influenced by the owner or veterinarian, they can be taken into consideration beforehand when selecting breeds.

Overall, some factors were identified that may help to reduce OE development. Thus, future studies should focus on protective features and further evaluate potential prophylactic measures to improve the quality of life for future generations of dogs.

## 6. Limitations

As this was a retrospective study conducted in a referral hospital, a certain bias regarding case load was unavoidable.

Another limiting factor was the use of weight in kilograms as widely used in this hospital to diagnose being overweight. The approach of using more dynamic systems like body condition scoring would lead to more accurate data.

Due to the purposeful selection of the control group from the Unit for Reproductive Medicine, a great portion of our study population was not neutered. However, percentual presentation of the data levels this limitation to a certain extent, as in our country, numerous dogs remain unneutered even without breeding purposes. Furthermore, a certain bias regarding breed distribution may be present, as depending on trends, certain breeds may be under- or overrepresented.

Lastly, the use of Bonferroni correction reduces the risk of a type I error but simultaneously increases type II errors. This may lead to false non-significant results. Thus, descriptive statistics were used when it was a near-miss of significance.

Regarding the statistical approach, multivariate analysis would be more fitting in medicine-related topics as diseases tend to be multifactorial. Through multivariate analysis, it is possible to analyze multiple factors at once and their influence on each other. Nonetheless, the purpose of this study was to evaluate the significance of certain factors individually. Thus, bivariate analysis was chosen and supplemented by further post hoc testing to ensure a legitimate statistical approach.

Despite these described limitations, this study found, for the first time, a significant increased risk for brachycephalic breeds regarding canine OE in a univariate analysis. Furthermore, this study showed tendencies of potential influences of body size and neuter status on developing OE.

## Figures and Tables

**Figure 1 animals-15-00933-f001:**
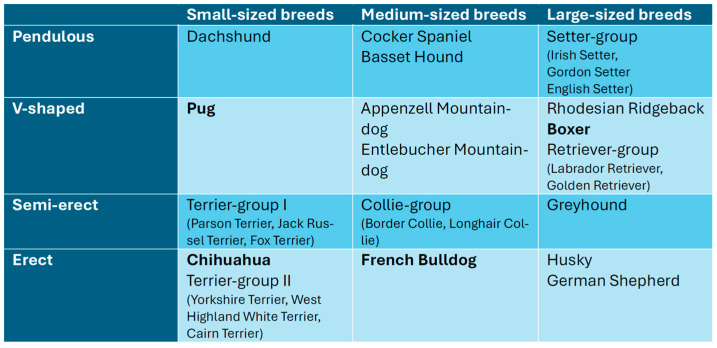
Table of all 18 dog breeds that were used in this study divided by their body size and pinna formation. Brachycephalic breeds are highlighted in bold.

**Figure 2 animals-15-00933-f002:**
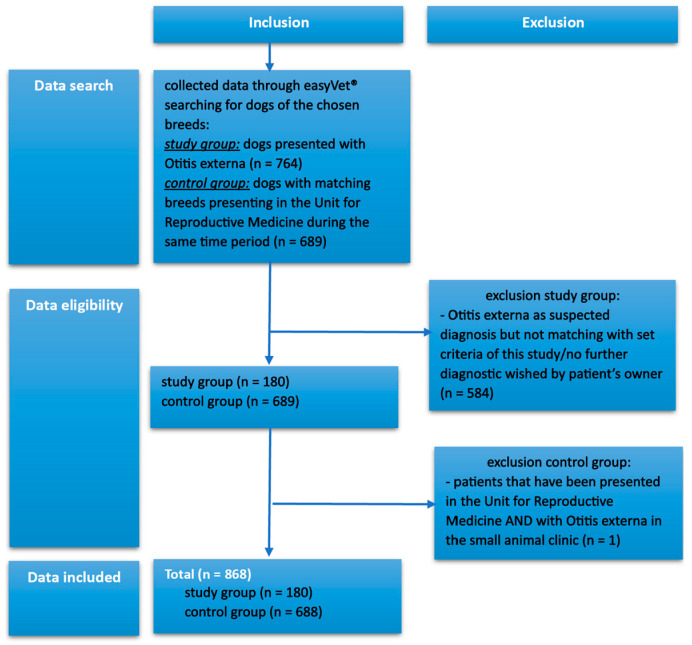
A detailed summary of the initial data’s selection process, leading to a total number of 868 cases included in this study. As seen above, one case was excluded as the dog was part of the control group but diagnosed with Otitis externa.

**Figure 3 animals-15-00933-f003:**
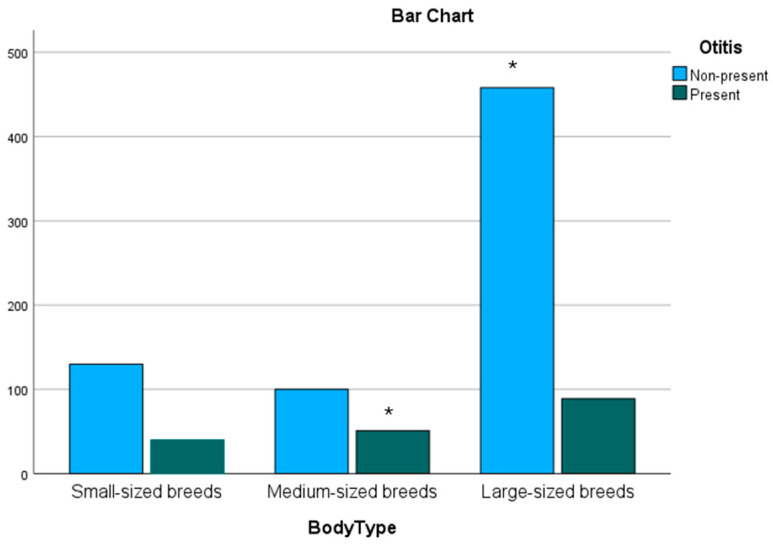
Bar chart showing the presence of Otitis externa grouped by body size. * Large-sized breeds are shown to have significantly more dogs in the non-present group, while medium-sized dogs have a significance regarding the presence of Otitis externa.

**Figure 4 animals-15-00933-f004:**
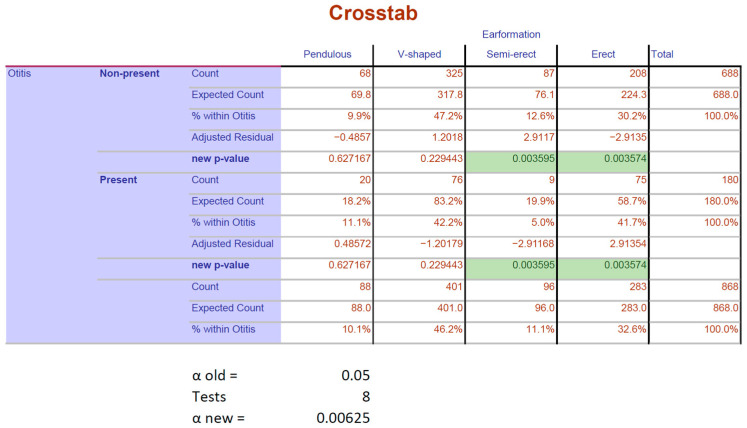
Results after post hoc testing by applying Bonferroni correction for the analysis of pinna formation. Both rows defined by “new *p*-value” contain results from post hoc testing based on the “Adjusted Residuals”. The green-highlighted cells indicate the values that remained significant after Bonferroni correction: semi-erect ears and erect ears.

**Figure 5 animals-15-00933-f005:**
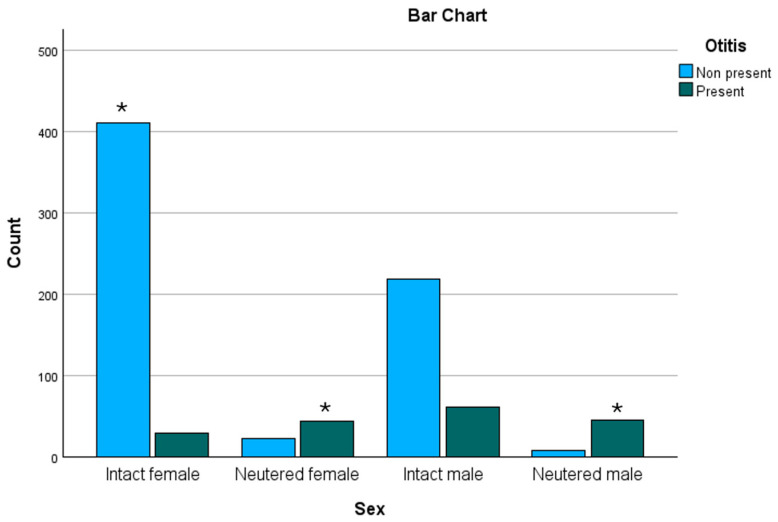
Bar chart showing the presence of Otitis externa grouped by sex and neuter status, bearing in mind that the control group was retrieved from the Unit of Reproductive Medicine. * Bars that are connected to significant results are highlighted, in this case, intact females with significance in the non-present, neutered females in the present, and neutered males in the present as well.

**Figure 6 animals-15-00933-f006:**
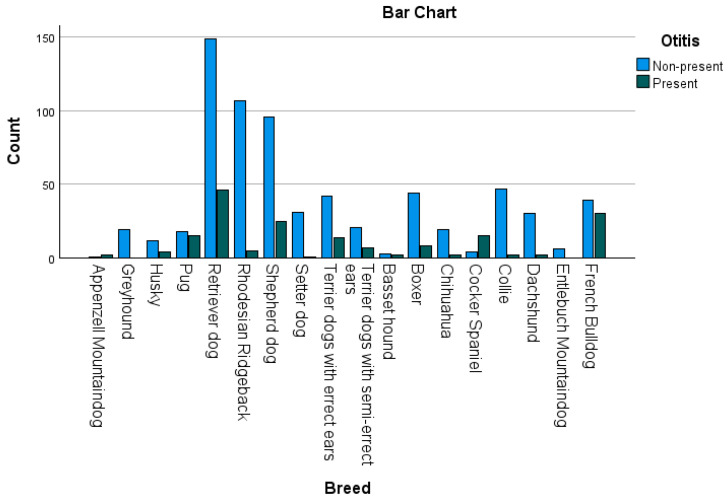
Bar chart of all breeds and groups divided by their presence of Otitis externa. It shows an overrepresentation of Retriever dogs, Ridgebacks, Shepherd dogs, and French Bulldogs. After post hoc testing, Rhodesian Ridgebacks showed a significantly low risk for Otitis externa while Pugs, French Bulldogs, and Cocker Spaniels had a significantly higher risk. It should be borne in mind that a high percentage of the dogs were retrieved from the Unit for Reproductive Medicine; thus, a certain degree of influence regarding popular dog breeds may exist.

**Figure 7 animals-15-00933-f007:**
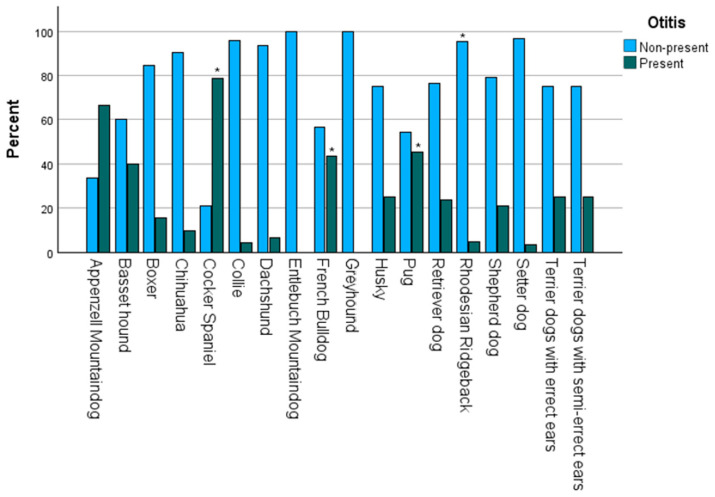
Bar Chart with the percentage distribution of the presence of Otitis externa grouped by breeds. * Bars that are connected to significant results are highlighted: in this case, Rhodesian Ridgeback with significance in the non-present group; Cocker Spaniel, French Bulldog, and Pug in the present.

**Figure 8 animals-15-00933-f008:**
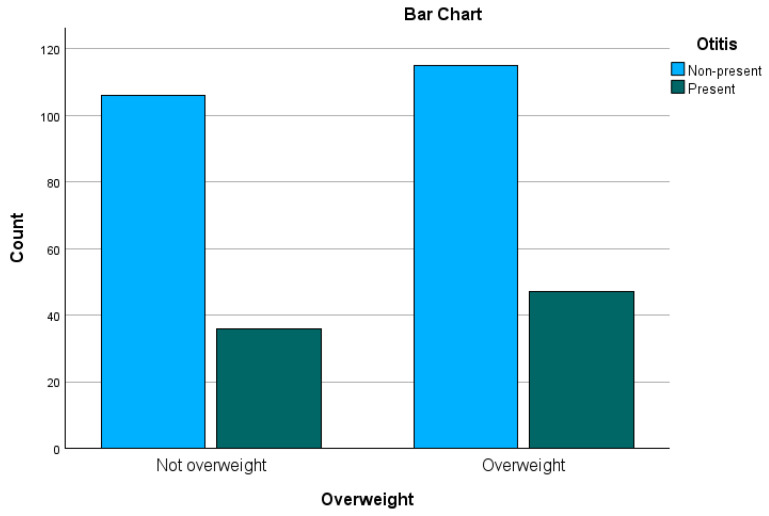
Bar chart of the distribution of overweight patients and those who are not overweight grouped by the presence of Otitis externa.

**Figure 9 animals-15-00933-f009:**
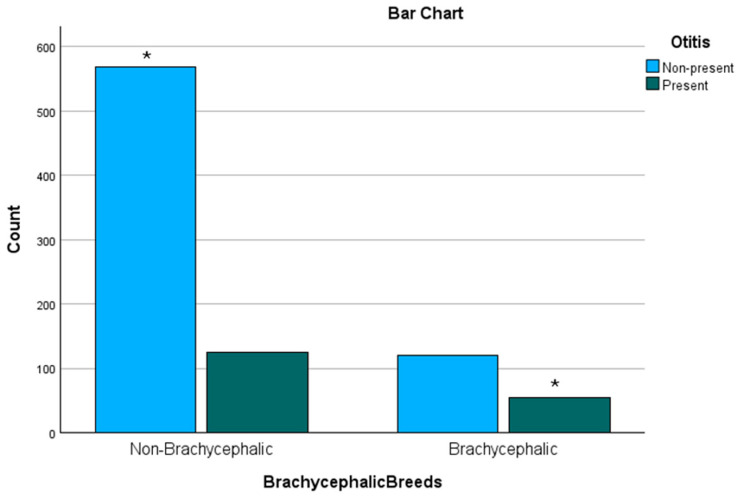
Bar chart of brachycephalic breeds grouped by the presence of Otitis externa. * While approximately one third of the brachycephalic breeds developed Otitis externa, non-brachycephalic breeds presented Otitis externa less frequently.

**Figure 10 animals-15-00933-f010:**
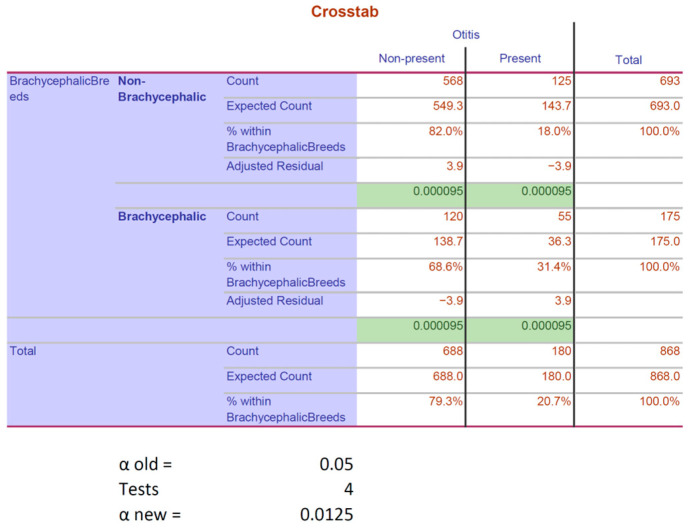
Results after post hoc testing by applying Bonferroni correction regarding brachycephaly. Both rows defined by “new *p*-value” contain results from the post hoc testing based on the “Adjusted Residuals”. The green-highlighted cells indicate the values that remained significant after Bonferroni correction. In this case, all values remained significant.

**Table 1 animals-15-00933-t001:** Chi-square test showing significance between Otitis externa and breeds (* df = degrees of freedom).

	Value	df *	Asymptotic Significance (2-Sided)
Pearson–Chi-Square	122.685	17	<0.001
No. of Valid Cases	868		

## Data Availability

The original data presented in the study are openly available on FigShare at https://doi.org/10.6084/m9.figshare.28324157.

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
