# Peer review of "Brachycephaly, Ear Anatomy, and Co—Does Size Matter? A Retrospective Study on the Influence of Size-Dependent Features Regarding Canine Otitis Externa"

_animals, 2025, doi:10.3390/ani15070933_

Round 1

Reviewer 1 Report

Comments and Suggestions for Authors

This was an interesting study with a large study population. The authors have used the data to investigate potential risk factors and protective factors for the development of OE.

It might be useful to use a program to check for correct grammar throughout the manuscript.

Line 14: please consider removing daily as this does not flow well grammatically.

Line 15:  Change to “Therefore, many studies have been conducted analyzing predispositions.”

Line 27 change “in our hospital” to “a referral small animal hospital.

Line 32 change develop to developing.

Line 34: remove were and give the comparison e.g. “presented more frequently with Otitis externa, than other dog breeds.”

Line 57 : change to size dependant.

Line 58: change to size dependant.

Line 60 change shape to morphology.

Line 69:  Please consider changing the flowing to the sentence below: In human medicine it  is described that overweight leads to an endocrine disease called “metabolic syndrome”.

“In the human field metabolic syndrome is an endocrine disease that is linked to being overweight/ obese.”

Line 71: “Although exceptions occur, this condition is mostly associated with obesity.”

Line 78 change: “had not been focused at” to “have not been previously investigated.”

Line 81 change “directing towards” to  “directed to”

Line 86: it would be interesting to classify which breeds were included in the brachycephalic group. Have you considered using the term extreme brachycephalic to describe Pugs, English Bulldogs and French Bulldogs?

Line 93: Please reconstruct this sentence as the meaning is not clear. “This decision regarding the control group should assure that none of these dogs had any significant health problems.”

Perhaps: The control group was sourced via the reproduction clinic to reduce the incidence of co-morbidities in this population.”

Was there an exclusion criteria for the control group?

Line 112 is ambiguous, do you mean some cocker spaniels were included in the brachycephalic group – please clarify this.

Figure 2 is a useful addition to explain the inclusion criteria. Please reference the reader to this when explaining the inclusion criteria for the control group.

Line 217-219 Please clarify if these breeds were found to have a statistically higher prevalence of OE.

Figure 4 and 10

Including the crosstab results does not add to the clarity of the statements. It would be preferable to write in a sentence what these results show or include a summary of the results with the adjusted p-values rather than the whole tables.

Line 259

Please include in the discussion here that this study actually did not find pendulous pinna conformation a risk factor for OE. My understanding is that semi-erect pinna was found to be lower risk or protective against development of OE and that erect ears had a higher risk of OE. This seems to be counter intuitive and warrants an explanation in the discussion.

Please clarify line 263/ 264 do you mean that there is bias in the control group as there are a high number of cocker spaniels (or another breed) presenting to the reproductive unit?

Line 268

Is the presence of French Bulldogs in the erect pinna group the reason that there was the finding of increased risk of OE in this pinnal morphology group?

I see that this was mentioned later in the discussion, please also mention this when discussing ear morphology.

Line 392 Please comment here if the control group influenced the breed distribution?

Comments on the Quality of English Language

Occasionally there were grammatical errors, in particular with regards to tense. This did affect the clarity of the document. 

Author Response

Comment 1: This was an interesting study with a large study population. The authors have used the data to investigate potential risk factors and protective factors for the development of OE. It might be useful to use a program to check for correct grammar throughout the manuscript.
Response 1: Thank you very much for this kind feedback! We used a program to scan the document and correct some english grammar errors. Of course a program is not fully perfect, so we also adressed a proof reader (which has an masters degree in german an english) but still waiting for feedback if they are available. As we do not want to miss the deadline we submit the current version but are more than willingly to provide another one if the English should habe to be improved further. 

Comment 2: Line 14: please consider removing daily as this does not flow well grammatically.
Response 2: Changed as stated above. 

Comment 3: Line 15:  Change to “Therefore, many studies have been conducted analyzing predispositions.”
Response 3: Changed as stated above. 

Comment 4: Line 27 change “in our hospital” to “a referral small animal hospital.
Response 4: Changed as stated above.

Comment 5: Line 32 change develop to developing.
Response 5: Changed as stated above.

Comment 6: Line 34: remove were and give the comparison e.g. “presented more frequently with Otitis externa, than other dog breeds.”
Response 6: Changed as stated above.

Comment 7: Line 57 : change to size dependant.
Response 8: Changed as stated above.

Comment 9: Line 58: change to size dependant.
Response 9: Changed as stated above.

Comment 10: Line 60 change shape to morphology.
Response 10: Changed as stated above.

Comment 11: Line 69:  Please consider changing the flowing to the sentence below: In human medicine it  is described that overweight leads to an endocrine disease called “metabolic syndrome”.
“In the human field metabolic syndrome is an endocrine disease that is linked to being overweight/ obese.”
Response 11: Thank you for this detailted suggestion. We changed it as stated above.

Comment 12: Line 71: “Although exceptions occur, this condition is mostly associated with obesity.”
Response 12: Changed as stated above.

Comment 13: Line 78 change: “had not been focused at” to “have not been previously investigated.”
Response 13: Changed as stated above.

Comment 14: Line 81 change “directing towards” to  “directed to”
Response 14: Changed as stated above.

Comment 15: Line 86: it would be interesting to classify which breeds were included in the brachycephalic group. Have you considered using the term extreme brachycephalic to describe Pugs, English Bulldogs and French Bulldogs?
Response 15: As wished, the innfo was added in the text with reference to Figure 1. The term "extreme brachycephaly" was discussed but finally dismissed to not prevent  further subcategorizing.

Comment 16: Line 93: Please reconstruct this sentence as the meaning is not clear. “This decision regarding the control group should assure that none of these dogs had any significant health problems.” Perhaps: The control group was sourced via the reproduction clinic to reduce the incidence of co-morbidities in this population.” Was there an exclusion criteria for the control group?
Response 16:  The sentenced was reconstructed according to the suggestion, thanks again for the detailed recommendation. Furthermore the exclusion criteria were included in the main text with a reference to Figure 2.

Comment 17: Line 112 is ambiguous, do you mean some cocker spaniels were included in the brachycephalic group – please clarify this.
Response 17: We reconstructed the setenced and hopefully clarified the issue.

Comment 18: Figure 2 is a useful addition to explain the inclusion criteria. Please reference the reader to this when explaining the inclusion criteria for the control group.
Response 18: Included a reference to Figure 2 in this part of the manuscript.

Comment 19: Line 217-219 Please clarify if these breeds were found to have a statistically higher prevalence of OE.
Response 19: We added all breeds with statistical significance. 

Comment 20: Figure 4 and 10 Including the crosstab results does not add to the clarity of the statements. It would be preferable to write in a sentence what these results show or include a summary of the results with the adjusted p-values rather than the whole tables.
Response 20: Thanks again for the feedback, we went along with an explanation of the results. If you however think a summary of the results would be more fitting, we would glady change that. 

Comment 21: Line 259 Please include in the discussion here that this study actually did not find pendulous pinna conformation a risk factor for OE. My understanding is that semi-erect pinna was found to be lower risk or protective against development of OE and that erect ears had a higher risk of OE. This seems to be counter intuitive and warrants an explanation in the discussion.
Response 21: Thank you for this commentary! You are right, it was not mentioned that pendulous ears did not have any significance and therefore a statement was added in the discussion. To not disturb the reading-flow regarding the breeds-paragraph, the entire pinna-formation-paragraph was attached directly to the one concerning breeds. Said paragraph includes potential explanations why erect ears may have an increased risk in this study (and also a potential influence of French Bulldogs). If there is any more discussion you may seem fit, we are more then willing to provide them. 

Comment 22: Please clarify line 263/ 264 do you mean that there is bias in the control group as there are a high number of cocker spaniels (or another breed) presenting to the reproductive unit?
Response 22: Added the clarification that the bias was meant regarding breeds and their representation in general. As the patients of the reproductive unit may vary depending on current trends in dog breeds and therefore may be over- or underrepresented. 

Comment 23: Line 268 Is the presence of French Bulldogs in the erect pinna group the reason that there was the finding of increased risk of OE in this pinnal morphology group? I see that this was mentioned later in the discussion, please also mention this when discussing ear morphology.
Response 23: Thank you again, as previously stated this whole paragraph was moved right next to discussion of ear morphology. 

Comment 24: Line 392 Please comment here if the control group influenced the breed distribution?
Response 24: Added potential distributio-bias to the "Limitations"-part of the manuscript. 

Reviewer 2 Report

Comments and Suggestions for Authors

The article prepared by Ponn et al. is an interesting approach to studies on canine otitis externa, which is an important issue in clinical practice. The manuscript quality is very good, I have only few comments listed below.

Line 27: The Number of animals should be stated in the abstract

Figure 3: Post hoc analysis indicated significance in medium and large breeds, but only large was indicated on the Figure. It would be better to also indicate on the figure if the difference is considered between OE and non-OE group or between sizes.

Figures 4 and 10: These would be clearer if presented as Tables, not figures.

Figure 5: Similar to the comment by figure 3, please indicate the significance with characters or connecting lines

Figure 7: Also, add highlight to those 4 breeds

Figure 9: The same issue as above

Line 366, 374: I would not say “potential protective influence”; this is rather lower susceptibility

Author Response

Comment 1: The article prepared by Ponn et al. is an interesting approach to studies on canine otitis externa, which is an important issue in clinical practice. The manuscript quality is very good, I have only few comments listed below.
Response 1: Thank you very much for this kind feedback! We hope, we succesfully adapted the manuscript to your comments.

Comment 2: Line 27: The Number of animals should be stated in the abstract
Response 2: We included the total amount in the abstract. 

Comment 3: Figure 3: Post hoc analysis indicated significance in medium and large breeds, but only large was indicated on the Figure. It would be better to also indicate on the figure if the difference is considered between OE and non-OE group or between sizes.
Response 3: We changed the figure along to this comment. An asterisk was added in all significant bars, indicating which part (OE or non-OE) was responsible for significance. 

Comment 4: Figures 4 and 10: These would be clearer if presented as Tables, not figures.
Response 4: Thank you for this feedback. We tried to implement tables with all the data which can be seen in the Figure. Unfortunately, the tables would be quite cramped with not much space between the numbers. This led to a quite chaotic and not well readable table and may disrupt the reading flow. Furthermore, it would not be possible to implement “new-alpha” without messing up the table.
Therefore, we would prefer to keep the Figures, however, if you prefer a table, we will gladly change that. A small explanation was added to the description of the Figure to help potential readers understand the Table more clearly.

Comment 5: Figure 5: Similar to the comment by figure 3, please indicate the significance with characters or connecting lines
Response 5: Done as stated above. 

Comment 6: Figure 7: Also, add highlight to those 4 breeds
Response 6: Done as stated above. 

Comment 7: Figure 9: The same issue as above
Response 7: Done as stated above. 

Comment 8: Line 366, 374: I would not say “potential protective influence”; this is rather lower susceptibility
Response 8: We changed the wording in both lines.